# Federated Learning of Large Models at the Edge via Principal Sub-Model Training

**Yue Niu, Saurav Prakash**[*]
University of Southern California
`yueniu, sauravpr@usc.edu`

**Souvik Kundu**
Intel AI Labs
`souvikk.kundu@intel.com`

**Sunwoo Lee**
Inha University
`sunwool@inha.ac.kr`

**Salman Avestimehr**
University of Southern California
`avestime@usc.edu`

## Abstract

Limited compute and communication capabilities of edge users create a significant bottleneck for federated learning (FL) of large models. We consider a realistic, but much less explored, cross-device FL setting in which no client has the capacity to train a full large model nor is willing to share any intermediate activations with the server. To this end, we present Principal Sub-Model (PriSM) training methodology, which leverages models' low-rank structure and kernel orthogonality to train sub-models in the orthogonal kernel space. More specifically, by applying singular value decomposition (SVD) to original kernels in the server model, PriSM first obtains a set of principal orthogonal kernels in which each one is weighed by its singular value. Thereafter, PriSM utilizes a novel sampling strategy that selects different subsets of the principal kernels independently to create sub-models for clients. Importantly, a kernel with a large singular value is assigned with a high sampling probability. Thus, each sub-model is a low-rank approximation of the full large model, and all clients together achieve the near full-model training. Our extensive evaluations on multiple datasets in resource-constrained settings show that PriSM can yield an improved performance of up to 10% compared to existing alternatives, with only around 20% sub-model training.

## 1   Introduction

Federated Learning (FL) is emerging as a popular paradigm for distributed and privacy-preserving machine learning as it allows local clients to perform ML optimization jointly without directly sharing local data [21, 15, 19]. Thus, it enables privacy protection on local data, and leverages distributed local training to attain a better global model. This creates opportunities for many edge devices rich in data to participate in the joint training without direct data sharing. For example, resource-limited smart home devices can train local vision or language models using private data, and achieve a server model that generalizes well to all users via federated learning [23].

Despite significant progress in FL in the recent past, several crucial challenges still remain when moving to the edge. In particular, limited computation and communication capacities prevent clients from learning large models for leveraging vast amounts of local data at the clients. This problem has attracted a lot of attention [8, 11, 30, 27, 9]. For example, [8, 11, 30] propose to assign clients with different subsets of server model depending on their available resources. However, these works have an underlying assumption that some of the clients have sufficient resources to train a nearly full large model. As a result, server model size is limited by the clients with maximum computation

---

[*]Yue Niu, Saurav Prakash equally contribute to this work.

36th Conference on Neural Information Processing Systems (NeurIPS 2022).

and communication capacities. To overcome resource constraints on clients, prior works such as [27, 9] change the training paradigm by splitting a model onto server and clients. The computational burden on the clients is therefore relieved as the dominant part of the burden is offloaded to the server. However, such a methodology requires sharing of intermediate activations and/or labels with the server, which directly leaks input information and potentially compromises privacy promises of FL.

To overcome the aforementioned limitations in prior works, we focus on an even more constrained and realistic setting at the edge, in which no client is capable of training a large model nor is willing to share any intermediate data and/or labels with the server. To this end, we propose Principal Sub-Model (PriSM) training to *allow each client to only train a small sub-model, while still enabling the server model to achieve comparable accuracy as the full-model training*. We exploit low-rank structure in models during the training, which is commonly used in reducing compute costs [16, 7]. However, naive low-rank approximation in FL [30], where all clients only train top-k kernels based on their capacities, incurs a notable accuracy drop, especially in very constrained settings. In Figure 1, we delve into this issue by showing the number of principal kernels required in the orthogonal space to accurately approximate each convolution layer in the first two *ResBlocks* in ResNet-18 [10] during FL training[2]. We observe that even at the end of the FL training, around half of the principal kernels are still needed to sufficiently approximate each convolution layer. We have similar find-

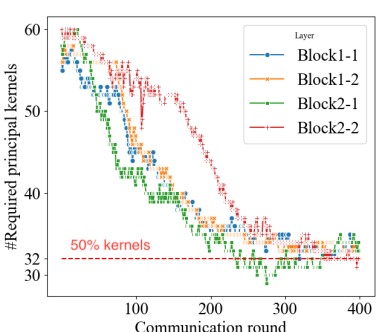

Figure 1: Number of principal kernels in the orthogonal space required to accurately approximate each of the two convolution layers in the first two ResBlocks in ResNet-18 during FL training. Block$i$-$j$ indicates $j$-th convolution layer in $i$-th ResBlock. Each of these convolution layers has 64 kernels in the original space.

ings for the remaining convolution layers. Therefore, to avoid the reduction in server model capacity, it is essential to ensure that all server-side principal kernels are collaboratively trained on clients, especially when each client can only train a very small sub-model (e.g., <50% of the server model).

Based on our above observations, PriSM employs a novel probabilistic strategy to select a subset of kernels and create a sub-model for each client as shown in Figure 2. More specifically, PriSM first converts the model into orthogonal space where original convolution kernels are decomposed into principal kernels using singular value decomposition (SVD). To approximate the original server model, PriSM utilizes our novel sampling process, that is based on the singular values, such that a principal kernel with a larger singular value has a higher sampling probability. Furthermore, the probabilistic process ensures that all sub-models can together provide a near server model coverage, thus leading to the near full-model training performance.

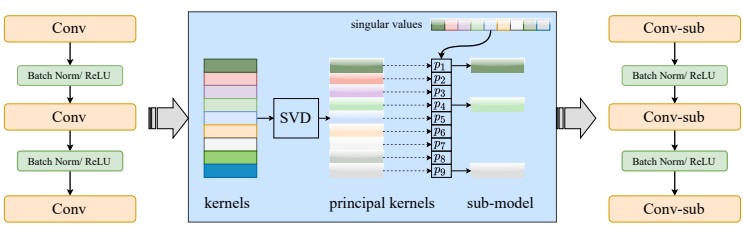

Figure 2: Creating clients' sub-models. PriSM randomly samples a subset of principal kernels to create a client's sub-model based on its computation and communication capacity. The sampling probability is derived from singular values of principal kernels. It ensures every sub-model approximates the full large model, and all sub-models together provide a near full server model coverage.

We conduct extensive evaluations for PriSM on vision and language tasks under resourced-constrained settings where no client is capable of training the large full model. Our results demonstrate that PriSM delivers consistently better performance compared to other prior works, especially when participating clients have very limited capacities. For instance, on ResNet-18/CIFAR-10, we show that PriSM only incurs around 2% and 3% accuracy drop for i.i.d and highly non-i.i.d datasets under a very

---

[2]See Sec B for further details, especially for calculating the required number of principal kernels.

constrained setting where all clients can only train around $20\%$ of the server model. Compared to other solutions, PriSM improves the accuracy by up to $10\%$.

## 2  Method

In this section, we first motivate our proposal, Principal random Sub-Model training (PriSM), with an observation of orthogonality in convolution layers. Then, we describe the details of PriSM.

**Notations.** We denote the Frobenius norm as $\|\cdot\|_F$, and $\sigma_i$ as $i$-th singular value in matrix $A$. $\circledast$ indicates convolution operation, and $\cdot$ indicates simple matrix multiplication. $\langle \cdot, \cdot \rangle$ denotes sum of element-wise multiplication or inner product. $tr(A)$ is the trace of matrix $A$.

### 2.1  Motivation: An Observation on Orthogonality

We consider a convolution layer with kernels $W \in \mathbb{R}^{N \times M \times k \times k}$ and input $X \in \mathbb{R}^{M \times H \times W}$, where $N$ and $M$ denote the number of output channels and the number of input channels, $k$ is kernel size, and $H \times W$ is the size of the input image along each channel. Based on a common technique *im2col* [5], the convolution layer can be converted to matrix multiplication as $\overline{Y} = \overline{W} \cdot \overline{X}$, where $\overline{W} \in \mathbb{R}^{N \times Mk^2}$ and $\overline{X} \in \mathbb{R}^{Mk^2 \times HW}$. For kernel decorrelation, we apply singular value decomposition (SVD) to map kernels into orthogonal space as: $\overline{W} = \sum_{i=1}^{N} \sigma_i \cdot \boldsymbol{u}_i \cdot \boldsymbol{v}_i^T$, where $\{\boldsymbol{u}_i\}_{i=1}^{N}$, $\{\boldsymbol{v}_i\}_{i=1}^{N}$ are two sets of orthogonal vectors[3]. The convolution can be decomposed as follows:

$$\overline{Y} = \sum_{i=1}^{N} \overline{Y}_i = \sum_{i=1}^{N} \sigma_i \cdot \boldsymbol{u}_i \cdot \boldsymbol{v}_i^T \cdot \overline{X}. \tag{1}$$

For $\forall i \neq j$, it is easy to verify that $\langle \overline{Y}_i, \overline{Y}_j \rangle = \sigma_i \cdot \sigma_j \cdot tr(\overline{X}^T \cdot \boldsymbol{v}_i \cdot \boldsymbol{u}_i^T \cdot \boldsymbol{u}_j \cdot \boldsymbol{v}_j^T \cdot \overline{X}) = 0$, namely the output features $\overline{Y}_i$ and $\overline{Y}_j$ are orthogonal. Therefore, if we regard $\overline{W}_i = \sigma_i \cdot \boldsymbol{u}_i \cdot \boldsymbol{v}_i^T$ as a principal kernel, different principal kernels create orthogonal output features. To illustrate this, Figure 3 shows a input image (left) and the outputs (right three) generated by principal kernels. We can observe that principal kernels captures different features and serve different purposes.

As [29, 3, 28] reveal, imposing orthogonality leads to better training performance. This motivates us to initiate the training with a set of orthogonal kernels. Furthermore, to preserve kernel orthogonality during training, it is critical to constantly refresh the orthogonal space through re-decomposition. The above intuitions play an important role in PriSM, which is described in the following section.

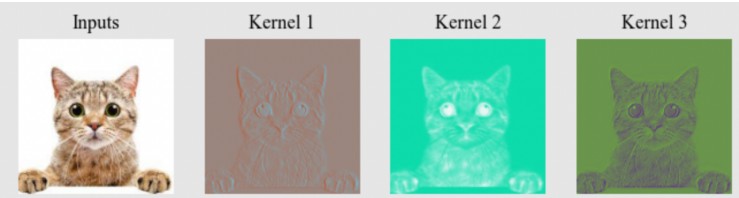

Figure 3: Orthogonal outputs generated by principal kernels $\overline{W}_i$. Different principal kernels capture different features: Kernel 1 extracts the outline of an object, Kernel 2 and 3 capture detailed textures of the object but on distinct regions.

### 2.2  PriSM: Principal Random Sub-Model Training

In the realistic setting considered in our paper, participating clients are very resource-limited and incapable of performing full large model training. Therefore, to train a large server model in FL, it is essential to distribute the training workloads among clients. One way toward this goal is to conduct sub-model training on clients. Based on the motivation in Section 2.1, we select sub-models and sample from the orthogonal space. Further, as we observe in Section 2.1, different principal kernels capture distinct features in the orthogonal space, and their contributions are further weighed by their corresponding singular values as shown in Eq. (1). Therefore, we propose a novel importance-aware

---

[3]We assume w.l.o.g $\overline{W}$ is a tall matrix.

sampling strategy to create sub-models for clients. This provides two key benefits: 1) each sub-model is a low-rank approximation to the server model; 2) the conglomerate of the sampled sub-models enables a near-full server model coverage. Each participating client trains its sub-model and uploads it to the server. The server then aggregates the updated sub-models, obtain the full model in the original space, and then re-decompose it to refresh the orthogonal space for the next round.

---

**Algorithm 1** PriSM: Principal Random Sub-Model Training

---

**Input:** layer parameters $W$, client capacities.
1: **for** communication round $t = 1, \cdots, T$ **do**
2:     Decompose $W$ into orthogonal kernel using SVD $\rightarrow \{\overline{W}_i\}_{i=1}^{N}$.
3:     Choose a subset of clients $\rightarrow \mathcal{C}$.
4:     **for** each client $c \in \mathcal{C}$ **do**
5:         Compute the sub-model size for client $c \rightarrow |\mathcal{I}_c|$.
6:         Obtain a sub-model using random sampling based on Eq. (2) $\rightarrow \mathcal{I}_c, \overline{W}^c$.
7:         Perform **LocalTrain** $\leftrightarrow \mathcal{I}_c, \overline{W}^c$. // Local training
8:     **end for**
9:     Aggregate parameters based on Eq. (3) $\overline{W} \leftarrow \{\overline{W}^c\}_{c \in \mathcal{C}}$. // Sub-model aggregation
10:     Reconstruct $W$ from $\overline{W}$. // Orthogonal space refresh
11: **end for**

---

12: **LocalTrain** $\leftrightarrow \mathcal{I}_c, \overline{W}^c$
13: **for** local iteration $k = 1, \cdots, K$ **do**
14:     Sample an input batch from the local dataset $\rightarrow \mathcal{D}_k$.
15:     Perform the forward and backward pass $\leftarrow \mathcal{D}_k, \overline{W}^c$.
16:     Compute additional gradients from regularization based on Eq. (4) $\leftarrow reg$.
17:     Update the local sub-model using SGD $\rightarrow \overline{W}^c$.
18: **end for**

---

We describe each components of PriSM below.

**Sub-model sampling.** Given a resource budget in the client, suppose it can at most process $r$ principal kernels for a convolution layer. For a convolution layer with principal kernels $\{\overline{W}_i\}_{i=1}^{N}$, and the corresponding singular values $\{\sigma_i\}_{i=1}^{N}$, we randomly sample $r$ principal kernels denoted by $\mathcal{I}_c$ without replacement with sampling probability for $i$-th kernel as follows:

$$p_i = \frac{\sigma_i^{\kappa}}{\sum_{j=1}^{N} \sigma_j^{\kappa}}. \tag{2}$$

Here, $\kappa$ in Eq. (2) is a smooth factor that controls the probability distribution for principal kernels to be chosen. Therefore, with our proposed stochastic sampling strategy, *important* kernels with large singular values are more likely to be chosen, and all sub-models together provide a near-full model coverage. Other element-wise layers such as ReLU and batch normalization remain the same.

**Local training.** On each client, when performing optimization, the selected $\{\boldsymbol{u}_i, \boldsymbol{v}_i\}_{i \in \mathcal{I}_c}$ are optimized, together with trainable parameters in other layers. While in PriSM, singular values $\{\sigma_i\}_{i \in \mathcal{I}_c}$ are not updated during local training. This is because each singular value indicates *importance* of each principal kernel. Thus, freezing singular values across clients helps maintain consistency regarding importance for kernels and is useful in the aggregation of the sub-models as described next.

**Sub-model aggregation.** On the server side, with sub-models obtained from clients, we first aggregate $i$-th principal kernel as follows:

$$\overline{W}_i = \sigma_i \cdot \left(\sum_{c \in \mathcal{C}} \alpha_i^c \boldsymbol{u}_i^c\right) \cdot \left(\sum_{c \in \mathcal{C}} \alpha_i^c \boldsymbol{v}_i^c\right)^T, \tag{3}$$

where $\mathcal{C}$ denotes the subset of active clients, $\alpha_i$ is the aggregation coefficient for $i$-th kernel. We propose a weighted averaging scheme: if $i$-th kernel is selected and trained by $C_i$ clients, then $\alpha_i^c = 1/C_i$. Furthermore, for principal kernels that are not sampled during creation of sub-models, they remain intact during aggregation.

**Orthogonal space refresh.** The full model in the original space is constructed by converting each 2-dimensional $\overline{W}_i$ to the original dimension $\mathbb{R}^{M \times k \times k}$ and combining them. For the next communication round, the orthogonal space is refreshed by decomposing the updated $\overline{W}$ using SVD.

We further use two additional techniques to improve learning efficiency in the orthogonal space: activation normalization, and regularization on orthogonal kernels.

**Activation normalization.** We apply batch normalization without tracking running statistics; namely, the normalization always uses current batch statistics in the training and evaluation phases. Each client applies normalization separately with no sharing of statistics during model aggregation. Such an adaptation is effective in ensuring consistent outputs between different sub-models and avoids potential privacy leakage through the running statistics [2]. Hence, it has been used in several sub-model FL training schemes [8, 30].

**Regularization.** When learning a sampled subset of principal kernels on a client, naively applying weight decay to $\boldsymbol{u}_i$ and $\boldsymbol{v}_i$ separately results in poor final accuracy. Inspired by [17], for training on client $c$, we add regularization to the subset of kernels as follows:

$$reg = \frac{\lambda}{2} \left\| \sum_{i \in \mathcal{I}_c} \sigma_i \cdot \boldsymbol{u}_i \cdot \boldsymbol{v}_i^T \right\|_F^2 , \tag{4}$$

where $\lambda$ is the regularization factor, $\mathcal{I}_c$ denotes the subset of principal kernels on client $c$.

Algorithm 1 presents an overall description of PriSM. We only show the procedure on a single convolution layer with kernels $W$ for the sake of simplifying notations.

In the following remarks, we differentiate PriSM from Dropout and Low-Rank compression.

*Remark* 2.1. **PriSM vs Dropout.** PriSM shares some computation similarity with model training using regular dropout [26]. However, regular dropout on the original kernels leads to significant convergence instability, especially with a high dropout probability [11]. In contrast, PriSM performs importance-aware sampling based on singular values. Therefore, each sub-model is an approximation to the full model, and training different sub-models on clients does not create significant inconsistency.

*Remark* 2.2. **PriSM vs Low-Rank Compression.** While PriSM exploits low-rank properties in models, it is not a low-rank compression method. Low-rank compression methods aim to construct a smaller server model by completely discarding some kernels even though they can still contribute to model performance. PriSM randomly select sub-models so that every kernel is possible to be learned.

# 3 Experiments

We evaluate PriSM under resourced-constrained settings where no clients can train the large full model. Specifically, we consider homogeneous settings with all clients having the same limited compute and communication capacity. We also compare PriSM with two other baselines: ordered kernel dropout in orthogonal space (OrthDrop), such as in [30]; and ordered kernel dropout in original space (OrigDrop), such as in [11, 8]. At a high level, our results demonstrate that PriSM achieves comparable server model accuracy even when only training very small sub-models on clients.

**Baselines.** Prior methods such as FjORD [11] and HeteroFL [8] select sub-models from the original kernel space, for which we denote as OrigDrop. On the other hand, we use OrthDrop to denote selecting fixed top-k principal kernels from the orthogonal space such as in FedHM [30].

**Models and Datasets.** We train ResNet-18 on CIFAR-10 [18], CNN on FEMNIST [4] and LSTM model on IMDB [20][4]. ResNet-18 is optimized for CIFAR-10, where kernel size in the first convolution layer is reduced to $3 \times 3$. CNN is a small model with two convolution layers.

**Data Distribution.** For CIFAR-10 and IMDB, we create balanced datasets during training with FL. Given the total number of samples and participating clients, we uniformly sample the equal number

---

[4]Due to the page limit, FEMNIST and LSTM results are deferred to Appendix A.2

of training images for each client when creating i.i.d datasets. For non-i.i.d datasets, we first use Dirichlet function $\text{Dir}(\alpha)$ [25] to create sampling probability for each client and then sample an equal number of training images for clients. We create two different non-i.i.d datasets with $\alpha = 1$ and $\alpha = 0.1$, where a smaller $\alpha$ denotes a higher degree of non-i.i.d. For FEMNIST [4], we directly use the dataset without any additional preprocessing.

**FL Setting.** We simulate an FL setting with 100 clients, and 20 clients are chosen uniformly at random in each communication round. Each client trains its model for 2 local epochs and then uploads it to the aggregation server. We use SGD with momentum in the local training. Further details are provided in Appendix A.1.2.

**Performance on homogeneous clients**. In this setting, we assume that all clients have the same limited compute and communication capacity. We vary the client sub-model size from 0.2 to 0.8 of the full server model, where $0.\times$ indicates only a $0.\times$ subset of the principal kernels are sampled in each convolution layer from the server model (denoted as *keep ratio* in the results). In Table 1, we list the computational and communication footprints for different sub-models of ResNet-18. For OrthDrop, we follow the strategy in [30] and select the top $0.\times$ principal kernels for all clients. For OrigDrop, we select the first $0.\times$ original kernels as in [8, 11].

Table 1: Model size and compute costs for different sub-models in PriSM.

| Model fraction | Full | 0.8 | 0.6 | 0.4 | 0.2 |
|---|---|---|---|---|---|
| ResNet-18 on CIFAR-10 | | | | | |
| Params | 11 M | 9.9 M (90%) | 7.4 M (67%) | 4.9 M (44%) | 2.5 M (22%) |
| MACs | 1.1 G | 0.9 G (80%) | 0.7 G (60%) | 0.5 G (45%) | 0.25 G (22%) |

Figure 4 shows final validation accuracy of ResNet-18 with different sampled sub-model sizes on i.i.d and non-i.i.d ($\alpha = 1, 0.1$) local datasets. We note that PriSM constantly delivers better performance than the other two baselines. The performance gap is even more striking under very constrained settings. For instance, when only $0.2\times$ sub-models are supported on clients, PriSM attains comparable accuracy as full-model training, and achieves up to $10\%$ performance improvement compared to OrthDrop on non-i.i.d dataset with $\alpha = 0.1$. Furthermore, we make two key observations. First, training with sub-models in the orthogonal space (OrthDrop) provides better performance than in the original space (OrigDrop), which aligns with our intuition in Section 2.1. Second, our importance-aware sampling strategy for creating sub-models is indispensable as demonstrated by the notable performance gap between PriSM and OrthDrop.

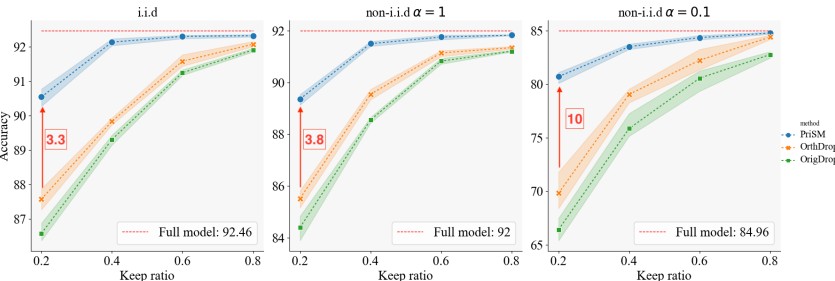

Figure 4: Training performance on CIFAR-10 on homogeneous clients. PriSM constantly delivers better performance compared to prior baselines, especially under very constrained settings.

## 4   Conclusion

We have considered the practical, yet under-explored, problem of federated learning in a resource-constrained edge setting, where no participating client has the capacity to train a large model. As our main contribution, we propose the PriSM training methodology, that empowers the resource-limited clients by enabling them to train smaller sub-models. At the same time, PriSM utilizes a novel sampling approach to obtain sub-models for the clients, all of which together ensure that the server

model achieves close to the full-model performance. Our extensive empirical results demonstrate that PriSM performs significantly better than the prior baselines, especially when each client can train only a very small sub-model. In particular, when each client is required to train a sub-model that is only around $20\%$ in size of the server model, we demonstrate that PriSM achieves a performance advantage of up to $10\%$ over the prior baselines.

## Acknowledgments and Disclosure of Funding

This material is based upon work supported by Defense Advanced Research Projects Agency (DARPA) under Contracts No. FASTNICS HR001120C0088 and HR001120C0160, NSF grants CCF-1763673, CNS-2002874, ARO grant W911NF-22-1-0165, and gifts from Intel, Qualcomm, and Cisco.

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

## A  Appendix

### A.1  Models and Hyperparameters

In this section, we provide detailed information about models and hyperparameter settings for the results presented in the paper. We will open-source our code upon acceptance of the paper.

### A.1.1 Models

**ResNet-18/CIFAR-10.** We use a ResNet-18 optimized for CIFAR-10, in which kernel size in the first convolution layer is changed from $7 \times 7$ to $3 \times 3$. Details are shown in Table 2.

Table 2: ResNet-18/CIFAR-10

| Module | | #kernels | size | stride | Batch Norm | ReLU | Downsample |
|---|---|---|---|---|---|---|---|
| Conv1 | | 64 | 3 | 1 | ✓ | ✓ | ✗ |
| ResBlock 1 | Block1-1 | 64 | 3 | 1 | ✓ | ✓ | ✗ |
| | Block1-2 | 64 | 3 | 1 | ✓ | ✓ | |
| ResBlock 2 | Block2-1 | 64 | 3 | 1 | ✓ | ✓ | ✗ |
| | Block2-2 | 64 | 3 | 1 | ✓ | ✓ | |
| ResBlock 3 | Block3-1 | 128 | 3 | 1 | ✓ | ✓ | ✓ |
| | Block3-2 | 128 | 3 | 1 | ✓ | ✓ | |
| ResBlock 4 | Block4-1 | 128 | 3 | 1 | ✓ | ✓ | ✗ |
| | Block4-2 | 128 | 3 | 1 | ✓ | ✓ | |
| ResBlock 5 | Block5-1 | 256 | 3 | 1 | ✓ | ✓ | ✓ |
| | Block5-2 | 256 | 3 | 1 | ✓ | ✓ | |
| ResBlock 6 | Block6-1 | 256 | 3 | 1 | ✓ | ✓ | ✗ |
| | Block6-2 | 256 | 3 | 1 | ✓ | ✓ | |
| ResBlock 7 | Block7-1 | 512 | 3 | 1 | ✓ | ✓ | ✓ |
| | Block7-2 | 512 | 3 | 1 | ✓ | ✓ | |
| ResBlock 8 | Block8-1 | 512 | 3 | 1 | ✓ | ✓ | ✗ |
| | Block8-2 | 512 | 3 | 1 | ✓ | ✓ | |
| Classification | | 10 | - | - | ✗ | ✗ | ✗ |

**CNN/FEMNIST.** We use a similar architecture as in FjORD [11]. The detailed model is shown in Table 3.

Table 3: CNN/FEMNIST

| Module | #kernels | size | stride | ReLU |
|---|---|---|---|---|
| Conv1 | 64 | 5 | 1 | ✓ |
| Pooling1 | - | 2 | 2 | ✗ |
| Conv12 | 64 | 3 | 1 | ✓ |
| Pooling2 | - | 2 | 2 | ✗ |
| Classification | 10 | - | - | ✗ |

**LSTM/IMDB.** We use a common LSTM model as shown in Table 4.

Table 4: LSTM/IMDB

| Module | input size | output size | hidden size | #layers |
|---|---|---|---|---|
| Embedding | 1001 | 64 | - | - |
| LSTM | 64 | 256 | 256 | 2 |
| FC | 256 | 1 | - | - |

### A.1.2 Training Hyperparameters

**ResNet-18/CIFAR-10 on homogeneous clients.** We simulate 100 clients during FL training, in which each client is assigned 500 training samples for both i.i.d and non-i.i.d datasets. In each

communication round, each client performs local training for 2 epochs using the local data, then uploads parameters to the server for aggregation. Table 5 lists detailed hyperparameters during FL training with ResNet-18.

Table 5: Hyperparameters for ResNet-18/CIFAR-10 on homogeneous clients

| Datasets | #clients | #samples | distribution | | augmentation |
|---|---|---|---|---|---|
| | 100 | 500 | i.i.d, non-i.i.d ($\alpha = 1, 0.1$) | | flip, random crop |
| Training | #Rounds | #local epochs | batch size | #active clients | smooth factor $\kappa$ |
| | 1000 | 2 | 32 | 20 | 2.5 |
| Optimization | Optimizer | Momentum | $wd$ | initial $lr$ | scheduler |
| | SGD | 0.9 | 0.0005 | 0.1 | cosine annealing |

**CNN/FEMNIST on homogeneous clients.** We simulate 100 clients during FL training, in which each client is assigned 10 users' data from the original training dataset. We use the whole validation dataset to compute the validation accuracy. Table 6 lists detailed hyperparameters during FL training with CNN.

Table 6: Hyperparameters for CNN/FEMNIST on homogeneous clients

| Datasets | #clients | #users | distribution | | augmentation |
|---|---|---|---|---|---|
| | 100 | 10 | natural non-i.i.d | | None |
| Training | #Rounds | #local epochs | batch size | #active clients | smooth factor $\kappa$ |
| | 300 | 2 | 32 | 20 | 2 |
| Optimization | Optimizer | Momentum | $wd$ | initial $lr$ | scheduler |
| | SGD | 0.9 | 0.0005 | 0.01 | cosine annealing |

**LSTM/IMDB on homogeneous clients.** During training, we simulate 100 clients, in which each client is assigned 375 training samples. We create local datasets with two different distributions using the same method as in CIFAR-10: i.i.d and non-i.i.d ($\alpha = 0.1$). Table 7 list detailed hyperparameters for training LSTM/IMDB.

Table 7: Hyperparameters for LSTM/IMDB on homogeneous clients

| Datasets | #clients | #samples | distribution | | augmentation |
|---|---|---|---|---|---|
| | 100 | 375 | i.i.d, non-i.i.d ($\alpha = 0.1$) | | None |
| Training | #Rounds | #local epochs | batch size | #active clients | smooth factor $\kappa$ |
| | 300 | 2 | 32 | 20 | 2 |
| Optimization | Optimizer | Momentum | $wd$ | initial $lr$ | scheduler |
| | SGD | 0.9 | 0.0002 | 0.1 | cosine annealing |

### A.2 Experiments on CNN/FEMNIST and LSTM/IMDB

Figure 5 and 6 show the training performance of PriSM on CNN/FEMNIST and LSTM/IMDB. Similar as in ResNet/CIFAR-10, PriSM consistently delivers the best final server model accuracy on both i.i.d and non-i.i.d datasets.

## B Model's Rank during Training

To analyze the server model's low-rank structure, we adopt a similar method as in [1] to calculate the required number of principal kernels to accurately approximate each layer as $2^{-\log\left(\sum_i p_i \log p_i\right)}$.

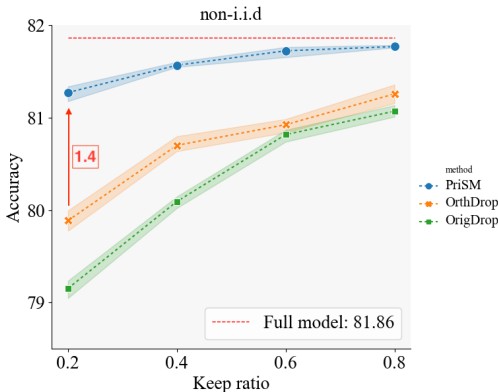

Figure 5: Training performance on FEMNIST on homogenous clients.

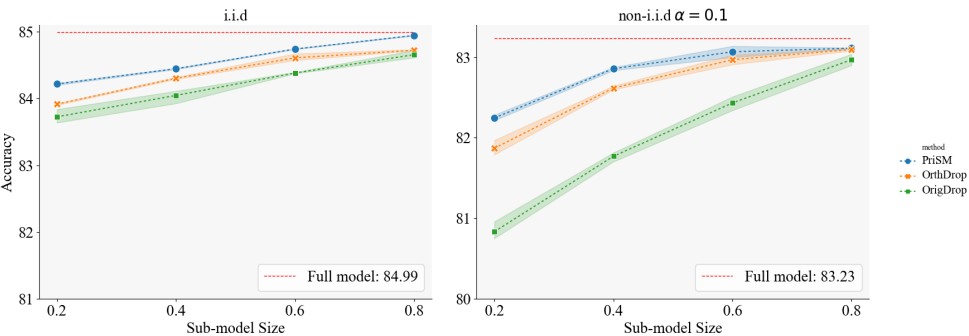

Figure 6: Training performance on IMDB on homogenous clients.

Here, $p_i$ is calculated as in Eq. (2) with $\kappa = 2$. Figure 7 shows the number of kernels required to approximate each layer with $3 \times 3$ kernels in ResNet-18 during FL with full models, where Block$i$-$j$ indicates $j$-th convolution layer in $i$-th ResBlock. First, we observe that while the server model attains a low-rank structure during training, a randomly initialized model does not exhibit a low-rank structure. Therefore, selecting a fixed set of top-k principal kernels for the sub-models inevitably causes reductions in the server model capacity. Furthermore, even at the end of the FL training, around half principal kernels are still required to approximate most layers. In fact, some layers require even more principal kernels. Therefore, our probabilistic sampling scheme is essential in preserving the server model capacity during FL training with sub-models.

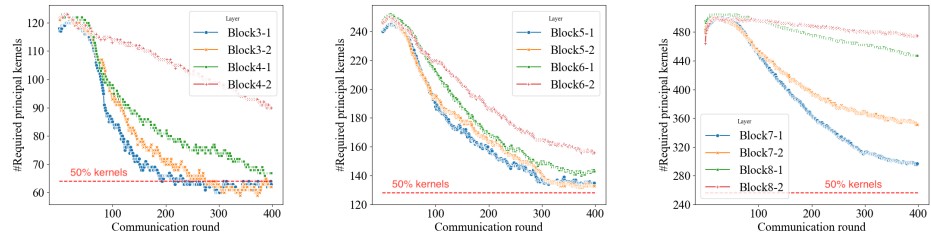

(a) ResBlock 3, 4 (128 kernels).  (b) ResBlock 5, 6 (256 kernels).  (c) ResBlock 7, 8 (512 kernels).

Figure 7: The number of principal kernels required to accurately approximate each convolution layer in ResBlocks 3-8 in ResNet-18 (Results of ResBlocks 1 and 2 are discussed in Figure 1).

## C  Related Works

**Factorized Models**. Training neural networks with layer factorization has been extensively studied in prior literature [7, 16, 14, 22, 13]. Specifically, these works are based on the observation that

well-trained neural networks have inherently low-rank structure and exhibit large-correlations across kernels. Hence, one can potentially down-size the model with a low-rank approximation to provide significant reduction in computations thus speeding up training. Furthermore, this can make model training more affordable for resource-constrained devices.

**Resource-Constrained Federated Learning**. While federated learning opens the door for collaborative model training over edge users having rich (but private) data, the computational and communication footprint prohibits training of large high-performance models at the resource-constrained clients. To address these resource limitations in federated learning, a number of works have been proposed in the literature [8, 11, 30, 8, 11, 30, 27, 24, 6, 9]. Particularly, in split learning [27, 24, 6], the model is partitioned into two parts, one (small) part is assigned to clients for local training, while the other (large) part is outsourced to the server. [9] proposes FedGKT that combines the model splitting approach with a novel bi-directional knowledge transfer technique between server and clients to achieve resource-constrained FL with much fewer communications than split learning. However, works such as split learning and FedGKT require sharing of intermediate activations (and in many cases, logits as well as labels) with the server, directly leaking input information and potentially compromising privacy promises of FL [31].

The works closely related to ours are HeteroFL [8], FjORD [11] and FedHM [30], that aim to enable participation of a resource-constrained client by letting it train a smaller sub-model based on its capabilities. In particular, HeteroFL and FjORD create sub-models for clients by selecting certain fixed number of original kernels of the server model. On the other hand, FedHM creates sub-models using fixed subsets of factorized principal kernels. However, in these works, the size of the server model gets limited by the clients with maximum computation and communication capacities, sacrificing the model performance. This becomes even more critical in a realistic, cross-device FL setting wherein no client has the capacity to train a large model. While another work, FedPara [12], proposes a low-rank factorized model training to reduce communication costs, computational footprint still remains prohibitive as every client is required to perform full-model training.

