# OpenReview forum: "Federated Learning of Large Models at the Edge via Principal Sub-Model Training"
_NeurIPS.cc/2022/Workshop/Federated_Learning — FL-NeurIPS 2022 Poster_

### Official Review · Reviewer_Xvtp · 2022-10-17
**Review of PriSM**

This paper considers a practical issue in Fl, i.e., communication bottlenecks of training large models, which has attracted much attention in this field. A new training method, called principal sub-model (PriSM), is proposed in this work by using SVD on the original data at the server side and then sampling the subsets to create the submodels for clients. It turns out that in many cases the proposed PriSM outperforms benchmarks, e.g.,FjORD and HeteroFL, w.r.t. the accuracies. The idea of this method is interesting and I believe that it should work well in the FL setting.
The major concern is that as it mainly focuses on large models, computing the kernels and performing SVD on these kernels would be costly, especially since these operations are needed at each communication round. If the server can afford this amount of computing and memory resources, why does it train the model?
Besides, there are many grammar mistakes and there is no theoretical justification regarding efficiency, convergence, generalization, etc. It is still not ready to convince the audience that this method can work well.

---

### Official Review · Reviewer_go38 · 2022-10-18
**Novel work with good contributions**

In this paper, the authors propose a framework called PriSM which uses only a few principle kernels in a layer and creates a sub-model, and performs Federated learning using that. The idea is nice, and the authors show good results with significant improvement from other compared methods.  This is certainly acceptable for a workshop paper, but the authors leave a lot of gaps making this reviewer eagerly wait for the full paper.

---

### Official Review · Reviewer_kHcd · 2022-10-18
**Interesting paper with promising results**

This paper introduces a new technique to train submodels for resource-constrained Federated Learning, named PriSM.

The authors build on prior works that proposed to train submodels to accommodate resource-constrained devices. The authors propose to use Atomo-like compression of kernels (https://proceedings.neurips.cc/paper/2018/file/33b3214d792caf311e1f00fd22b392c5-Paper.pdf) before communicating to clients. I think this is an essential reference that the authors need to discuss.

Overall, I think the paper is well written, and the provided results are promising. However, the paper does not introduce new ideas or novel combinations. In conclusion, I believe that despite the lack of novelty, the paper is still a good fit and relevant for the workshop audience.

Question for the authors:
1. How does your subsampling affect the theoretical convergence of FedAvg?
2. Does your sampling introduce any bias in the model?
3. Could you please showcase that training a bigger model than the one that the most resourceful client can accommodate can be beneficial?

---

### Decision · Program_Chairs · 2022-10-20

Accept (Poster)